# Mapping microbial ecosystems and spoilage-gene flow in breweries highlights patterns of contamination and resistance

Nicholas A Bokulich[1,2,3†], Jordyn Bergsveinson[4], Barry Ziola[4], David A Mills[1,2,3]*

[1]Department of Food Science and Technology, University of California, Davis, Davis, United States; [2]Department of Viticulture and Enology, University of California, Davis, Davis, United States; [3]Foods for Health Institute, University of California, Davis, Davis, United States; [4]Department of Pathology and Laboratory Medicine, University of Saskatchewan, Saskatoon, Canada

**Abstract** Distinct microbial ecosystems have evolved to meet the challenges of indoor environments, shaping the microbial communities that interact most with modern human activities. Microbial transmission in food-processing facilities has an enormous impact on the qualities and healthfulness of foods, beneficially or detrimentally interacting with food products. To explore modes of microbial transmission and spoilage-gene frequency in a commercial food-production scenario, we profiled hop-resistance gene frequencies and bacterial and fungal communities in a brewery. We employed a Bayesian approach for predicting routes of contamination, revealing critical control points for microbial management. Physically mapping microbial populations over time illustrates patterns of dispersal and identifies potential contaminant reservoirs within this environment. Habitual exposure to beer is associated with increased abundance of spoilage genes, predicting greater contamination risk. Elucidating the genetic landscapes of indoor environments poses important practical implications for food-production systems and these concepts are translatable to other built environments.

*For correspondence: damills@ucdavis.edu

Present address: †Department of Medicine, New York University Langone Medical Center, New York, United States

Competing interests: The authors declare that no competing interests exist.

## Introduction

Microbial activity is an inherent feature of food-processing systems, influencing the quality and healthfulness of foods for human consumption. Like other indoor environments, the building materials, substrates, and physiochemical conditions encountered by microbes in food-processing facilities differ dramatically from the outdoor conditions to which microbial life evolved (*Kelley and Gilbert, 2013*). How this has impacted the adaptation and ecological assemblages of microbes in food systems is largely unexplored. While the microbial communities of other indoor environments, such as homes (*Lax et al., 2014*), office buildings (*Kembel et al., 2014*), and hospitals (*Bokulich et al., 2013a*), can influence the health of their inhabitants, food-production facilities represent a unique type of built environment wherein microbial activities are intimately tied to product-quality outcomes. Thus, the sensory and safety effects of microbial growth in food-production streams have a much broader impact, such that consumer enjoyment and health can be linked to the hygiene and processing decisions at the food facility. Furthermore, both beneficial and detrimental microbial activities are well defined in foods, making these systems a useful model for exploring microbial ecosystem dynamics beyond the scope of contaminant mitigation.

Fermented foods, including beer, have the additional distinction that microbial activity is central to their production, responsible for necessary transformations as well as product spoilage (*Bokulich et al., 2012a*; *Bokulich and Bamforth, 2013*). Most modern food-fermentation practices

**eLife digest** Many microbes—including bacteria and fungi—can affect the food and drink we consume, for better and for worse. Some spoil food, making it less tasty or even harmful to health. However, microbes can also be important ingredients: for example, yeast ferments malted barley sugars to make the alcohol and flavor of beer.

Nowadays, many beers are made under carefully controlled conditions, where the only microbes in the beer should be the strain of yeast added to the barley sugars. A more traditional 'coolship' method can be used to make sour beers; the barley sugars cool in an open-topped vessel and are fermented by the yeast and bacteria found naturally on the raw ingredients and in the surrounding environment.

Relatively little was known about how microbes spread around and adapt to living inside buildings. Now, Bokulich et al. have used a range of molecular and statistical techniques to examine how bacteria and fungi are dispersed throughout a North American brewery that produces beer using both conventional and coolship brewing techniques. Most of the microbes found in the building originated from the raw ingredients used to make the beer, with different parts of the brewery containing different species. Over the course of a year, some species spread to new parts of the building; a statistical method predicted the sources of these microbes, and revealed some key areas and features of the brewery that affect microbial transfer.

Bokulich et al. also looked at the spread of genes that enable their bacterial hosts to spoil beer, including those that protect bacteria from the antimicrobial action of the hops that flavor many beers. Lactic acid bacteria are the main cause of beer spoilage and so are usually to be avoided in breweries, but are also a normal ingredient in sour beer. In the brewery Bokulich et al. investigated, beer-spoilage and hop-resistance genes were found throughout the brewery, even in areas not used to produce sour beer. However, little beer spoilage occurred.

The techniques used by Bokulich et al. to track the spread of microbes and their detrimental genes could be used in the future to understand how microbes adapt to other indoor environments. Indeed, Bokulich et al. suggest that breweries could be used as models to safely understand the factors that influence microbial movement in any food-production facility as well as other building environments.

occur under relatively aseptic conditions, employing pure starter cultures in their production, and environmental contamination represents a prevalent threat to product integrity (*Bokulich and Bamforth, 2013*). Beer is typically produced through fermentation of malted barley sugars (wort) to alcohol by pure strains of *Saccharomyces cerevisiae* or *Saccharomyces pastorianus*, and any additional organisms—including cross-contamination from different *Saccharomyces* strains used in the same facility—are considered contaminants (*Bokulich and Bamforth, 2013*). In traditional fermentation production practices, conversely, microbial communities introduced from raw materials and processing environments are central to the fermentation process. For example, in the production of coolship (lambic-style) beers, a type of sour beer, no starter cultures are added to the wort; instead, boiled wort is allowed to cool overnight in a shallow, open-top vessel known as a coolship where indigenous microbiota are presumably introduced to the product, initiating fermentation (*Van Oevelen et al., 1977*; *Bokulich et al., 2012b*; *Spitaels et al., 2014*). The unique succession of indigenous microbiota in coolship ale fermentations sets them apart chemically and sensorially from other beers (*Van Oevelen et al., 1976*; *Spaepen et al., 1978*), making coolship breweries a particularly interesting system for tracking microbial populations in food-processing environments.

Microbial spoilage genes are also well defined in brewing environments and can be studied in situ without representing a direct threat to public health. Beer is protected from wholesale microbial contamination through its alcoholic, low-pH, and antimicrobial properties, and modern sanitary technologies and practices minimize the threat of spoilage organisms that have evolved specifically to grow in beer (*Bokulich and Bamforth, 2013*). Nevertheless, biofilms and other environmental reservoirs remain potential sources of microbial contamination in breweries (*Timke et al., 2005*; *Storgards et al., 2006*; *Timke et al., 2008*; *Mamvura et al., 2011*; *Matoulkova et al., 2012*). As wort is boiled prior to fermentation, the primary reservoir for spoilage microorganisms in beer production

is the brewery environment. Lactic acid bacteria (LAB) are of particular concern, as some members of this clade resist hop antimicrobial compounds, enabling growth and spoilage of beer through acidification, hazes, and off-flavors (*Suzuki et al., 2006*; *Bokulich and Bamforth, 2013*). Hop iso-alpha-acids, the primary antimicrobial compounds in beer, kill most Gram-positive bacteria by acting as ionophores, dissipating proton-motive force across cell walls (*Simpson and Fernandez, 1992*, *1994*; *Simpson, 1993a*), and via oxidative stress (*Behr and Vogel, 2010*). Very few resistance mechanisms have been proposed (*Suzuki et al., 2006*), primarily being the multi-drug transporters *horA* (*Sami et al., 1997*; *Sakamoto et al., 2001*) and *horC* (*Suzuki et al., 2005*; *Iijima et al., 2006*), the transcriptional regulator of *horC*, *horB* (*Suzuki et al., 2005*; *Iijima et al., 2006*), and *hitA*, a divalent cation transporter (*Hayashi et al., 2001*) that imports manganese into the cell. These genes are all located on plasmids and transmit via horizontal gene transfer (*Suzuki et al., 2005*, *2006*). The frequency and transmission of these and other beer-spoilage genes within processing environments—or any other reservoir for spoilage microbes—have yet to be tested.

Here, we employ mixed molecular approaches (*Bokulich and Mills, 2012b*) and a Bayesian modeling method to interrogate the seasonal sources, reservoirs, and transmission of bacteria, fungi, and beer-spoilage genes within a North American brewery over the course of one year. Given the inherent role of environmental microbiota in conducting coolship ale fermentations, this serves as a model system for studying mechanisms for microbial transfer within food-processing systems. Information on population and gene flow extends to other food-processing systems where environmental microbiota are involved positively or negatively in the production, stability, and safety of human nutrition.

## Results and discussion

The microbial ecosystems of breweries are inherently linked to successful product outcomes, impacting the qualities and healthfulness of beer. In conventional brewing, any microbe not intentionally inoculated is a contaminant (*Bokulich and Bamforth, 2013*). Conversely, in coolship ale and other sour beer brewing, adventitious microbiota are integral to the process. Thus, we studied microbial ecosystem dynamics in a brewery that produces conventional, sour, and coolship beers, in order to observe food–ecosystem interactions from a dual perspective. First, how do microbial community assemblages and spoilage-gene frequencies change over time with respect to contamination issues in a conventional brewery? Second, how do these same elements interact with the production of sour beers?

### Production environment microbiota are driven by substrate contact

Short-amplicon marker-gene sequencing was employed to survey the bacterial and fungal consortia inhabiting the entire brewery environment. A total of 501 samples were collected during three seasons, representing the main processing surfaces and equipment used throughout the brewing process (*Figure 1*). Beta-diversity (between sample) comparisons provide useful assessments of the taxonomic similarity between different sites. Bray–Curtis dissimilarity of complete microbial profiles reveals that many samples cluster by processing room and substrate type regardless of season (*Figures 2–3*). Thus, fermenter samples cluster, associated with *Bacillaceae*; cellar production areas, associated with *Micrococcaceae (including the beer-spoiling genera Kocuria and Micrococcus)*; wort, malt, and hotside (wort-preparation) surfaces, associated with *Enterobacteriaceae, Leuconostocaceae, Candida santamariae, Pichia, and Rhodotorula*; barrel-room floor samples, associated with *Staphylococcaceae* and *Carnobacteriaceae*; and beer samples, associated with *Lactobacillaceae* and *Enterobacteriaceae*. Barrels cluster, associated with *Aspergillus, Eurotium*, and *Penicillium*; coolship and barrel-room samples with *Cryptococcus* and *Cladosporium*. These taxonomic trends each demonstrate significant site associations (Kruskal–Wallis Bonferroni-corrected $p < 0.05$). *S. cerevisiae* was common throughout the brewery, but especially in the fermentation cellar. LAB and acetic acid bacteria were found sporadically at different sites and times, including on and near packaging equipment and fermenters inoculated with LAB (*Figure 3*).

To further examine this relationship, the Bayesian technique sourcetracker (*Knights et al., 2010*) was used to test whether raw substrates may act as sources for the microbial consortia of brewery surfaces. This tool predicts the relative proportion of contamination in sink samples (in this case brewery surfaces) from microbial sources (raw ingredients and extraneous sources). Raw substrates (grain, hops, yeast, beer) and extraneous sources (human skin, outdoor air, soil, saliva, feces, freshwater,

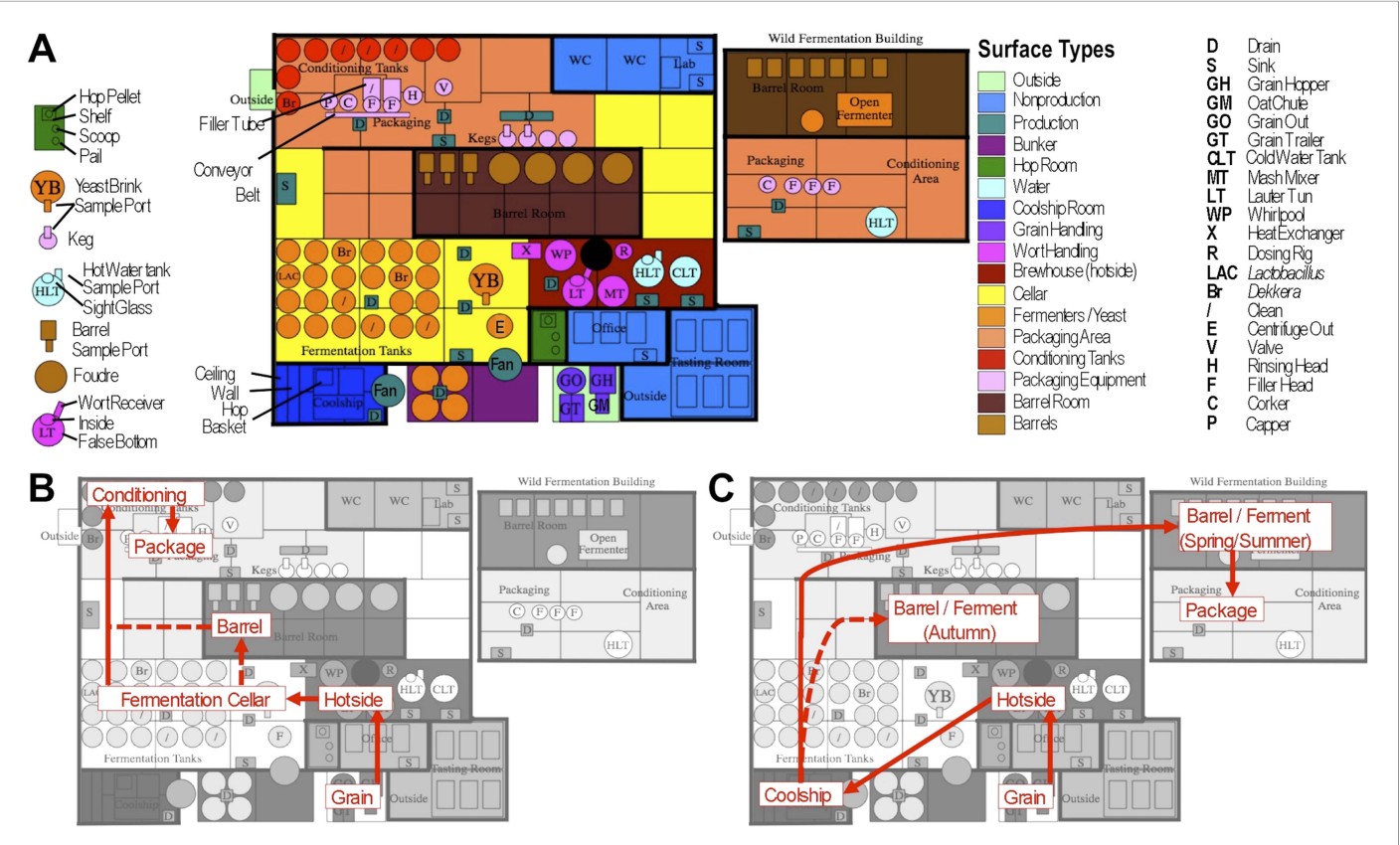

**Figure 1**. Brewery map and simplified brewing process diagrams. (**A**) Floorplan of brewery details surface sampling key and indicates separate sections of the brewery. LAC/Br fermenters indicate they were inoculated intentionally with lactic acid bacteria or *Dekkera* spp., respectively, at the time of sampling. (**B**) Process diagram for conventional beer brewing, illustrating the relationship between brewing stages and sections of the brewery. Grain is milled and taken to the brewhouse (hotside) area where it is mashed (steeped in hot water) to form wort, which is lautered (extracted from the grain by filtering and spraying with hot water) and then boiled with hops. Boiled wort is cooled and pumped to the fermentation cellar where it is inoculated with Saccharomyces and fermented. Optionally, barrel-aged beers are transferred to barrels after fermentation. Finished beers are transferred to conditioning tanks in a separate section of the brewery where they are cooled, carbonated, and then packaged. (**C**) Process diagram for coolship beer brewing. Same as conventional, but following boiling wort is pumped to the coolship room where it is left to cool overnight, exposed to the atmosphere. The following morning, the wort is pumped to barrels in which it is fermented and aged for 1–3 years. In the Autumn samples, this occurred in the barrel room in the main brewery, but in Spring and Summer this moved to a newly built facility dedicated to sour beers. All coolship and sour beers were packaged on separate equipment in this second facility. The distinction between coolship beers and sour beers is the use of this coolship; other sour beers are produced using conventional brewing methods (panel **B**), but are fermented with organisms other than Saccharomyces yeasts.

ocean water) from previous studies (*Caporaso et al., 2011*; *Bowers et al., 2012*; *Caporaso et al., 2012*; *Bowers et al., 2013*) were tested as microbial sources. Results reveal distinct patterns of contamination across seasons (*Figure 4*). Grains were predicted as the largest microbial contributor to hotside areas, almost all surfaces in the coolship room, and areas of the cellar away from the main fermentation area. Hops were predicted as the major contributor to cellar fermentation areas and fermentation equipment. Yeast was predicted as the highest contributor to fermenters, conditioning tanks, and packaging equipment. Beer was predicted to be a common contaminant around fermenters and barrels within the cellar. Skin was a minor contributor to some surfaces in the cellar. Other surfaces, including most barrels and barrel-room surfaces, were most influenced by unknown sources. Outdoor air, soil, saliva, feces, freshwater, and ocean water were predicted to provide only a very low level of contamination (<0.001 relative abundance). These results suggest that raw substrates are the main contaminant sources within the brewery, compared to extraneous sources. However, it is important to note that these predictions do not indicate a causative role for contamination. These predicted source/sink relationships could alternatively suggest that both are actually contaminated by another, untested source (e.g, fruit flies or other vectors could transfer microbes between these and other surfaces

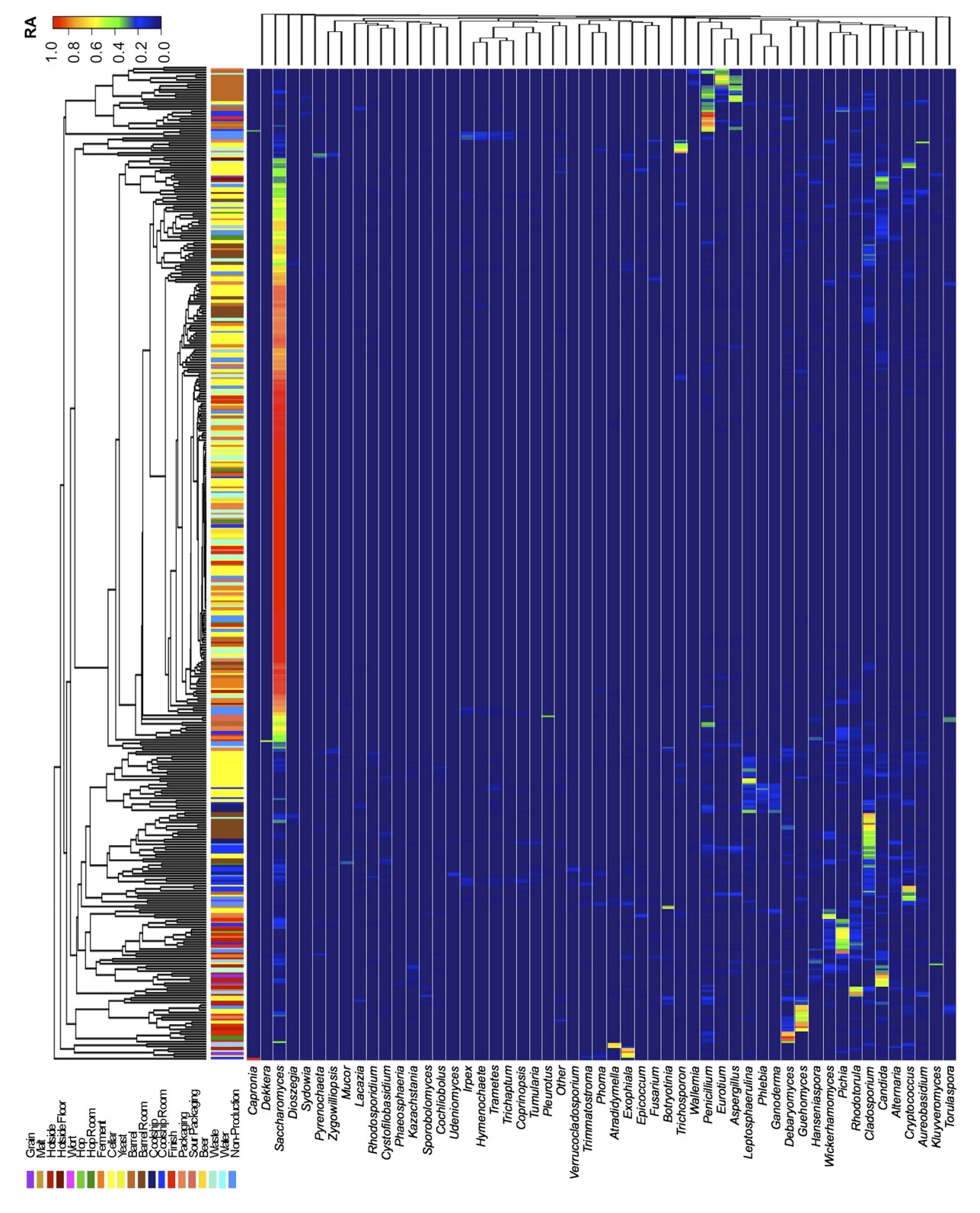

**Figure 2**. Taxon abundance heatmaps depict genus-level relative abundance of fungi across sampling sites detected by marker-gene sequencing. The relative abundances (RA) of each genus (columns) within each sample (rows) are indicated by the color of the intersecting tile. Sample types are indicated by colored bars to the left of each row, classified according to the location within the brewery (*Figure 1*) or the type of substrate (grain, wort, hops, beer).
*Figure 2. continued on next page*

Figure 2. Continued

Dendrograms represent Bray–Curtis dissimilarity between samples (vertical trees) and shared-niche similarity between taxa (horizontal trees), respectively indicating taxonomic composition similarities and taxon co-occurrence patterns. Only taxa ≥0.05 relative abundance in at least one sample are shown.

within the brewery). Nevertheless, these predictions highlight potential sources of contamination, or at least shared microbial transmission patterns, between substrates and equipment/surfaces within the brewery.

These findings suggest that substrate contact drives the microbial communities of different brewery surfaces. Consequently, raw materials (grain, hops, yeast) may be important vectors for any spoilage organisms encountered in the facility. This substrate-centric community structuring has also been observed in other food facilities (*Bokulich and Mills, 2013a*; *Bokulich et al., 2013b*, *2014*). However, the brewery surface niches clustered throughout time with only subtle changes across seasons, unlike wineries, which were significantly impacted by seasonal factors (*Bokulich et al., 2013b*). This reflects the divergent production schemes of these two foods: whereas most beers are produced year-round, wine is an inherently seasonal product. The raw materials for beer are stored and processed throughout the year in a continual production schedule, creating a stable environment on equipment surfaces that interact with these same raw substrates on a daily basis. Instead, brewery environments may function more like cheese-making environments, where facility-specific 'house' microbial communities form on equipment surfaces in response to idiosyncrasies in the indoor environment (*Bokulich and Mills, 2013a*). Such a possibility would have interesting implications for sour beer breweries, and comparative studies of lambic and coolship breweries could offer insight into the brewery-specific flavor profiles displayed in these beers.

## Physical mapping illustrates microbial dispersal in processing facility

By mapping surface samples to their physical location within the brewery, a spatial model emerges of microbial dispersion across brewery surfaces over time (*Figures 5–7*). Several populations visibly spread with time, most prominently *S. cerevisiae*, which progressed from dominating fermentation and packaging areas only in Autumn to being the most abundant fungus detected across the brewery in Spring and Summer. Likewise, *C. santamariae* displayed high abundance on hotside surfaces in Autumn but became more abundant and spread to nearby sections of the cellar in Spring and Summer (*Figure 5*). As building measurements were not taken during these times, the factors driving these changes cannot be assessed with these data, but warming temperatures and increasing humidity in the Spring and Summer months could be associated phenomena. Other taxa demonstrated more localized patterns of dispersion, such as *Micrococcus* and *Kocuria*, which appeared to spread the most around floors and other surfaces in cellar, barrel room, and packaging areas (*Figure 6*). Here, regular contact with beer runoff diluted with rinse water may support growth of these populations, which are associated with spoilage in low-alcohol beers (*Back, 1981*). High populations of *Acetobacter* and *Lactobacillus* were found more disparately, and specifically in areas where high volumes of wort and beer may be encountered: on conveyor belts and floors below packaging areas, hotside and cellar area sinks, and on sample ports for isolated fermenters and kegs (*Figure 6*). On these sites microbial communities can contact undiluted beer, for example, drips in the basin below the packaging-line belt, wort and beer collected for specific gravity (sugar concentration) measurements then dumped in sinks, and sampling ports on fermenters. This follows the known behavior of these bacteria, which can spoil undiluted, higher-alcohol beers under appropriate conditions (*Bokulich and Bamforth, 2013*).

These results illustrate the progressive dissemination of microbes in space and time within a functioning food-processing environment. Microbes not only associate with specific substrates, they exhibit patterns of dispersion within confined regions of the brewery. This yields useful insight into the transmission behavior of these organisms, and especially taxa associated with beer spoilage, through physical space. Microbial confinement within discrete zones suggests that physical barriers (e.g., walls) and physiochemical conditions (e.g., humidity and temperature control) can staunch the spread of some microbes. Non-production surfaces that encounter beer and/or waste streams, such as floors, sinks, and grain-handling equipment, are typically only cleaned by hose and may accumulate substrates for supporting microbial growth and survival. Aerosols and splashing occur regularly in

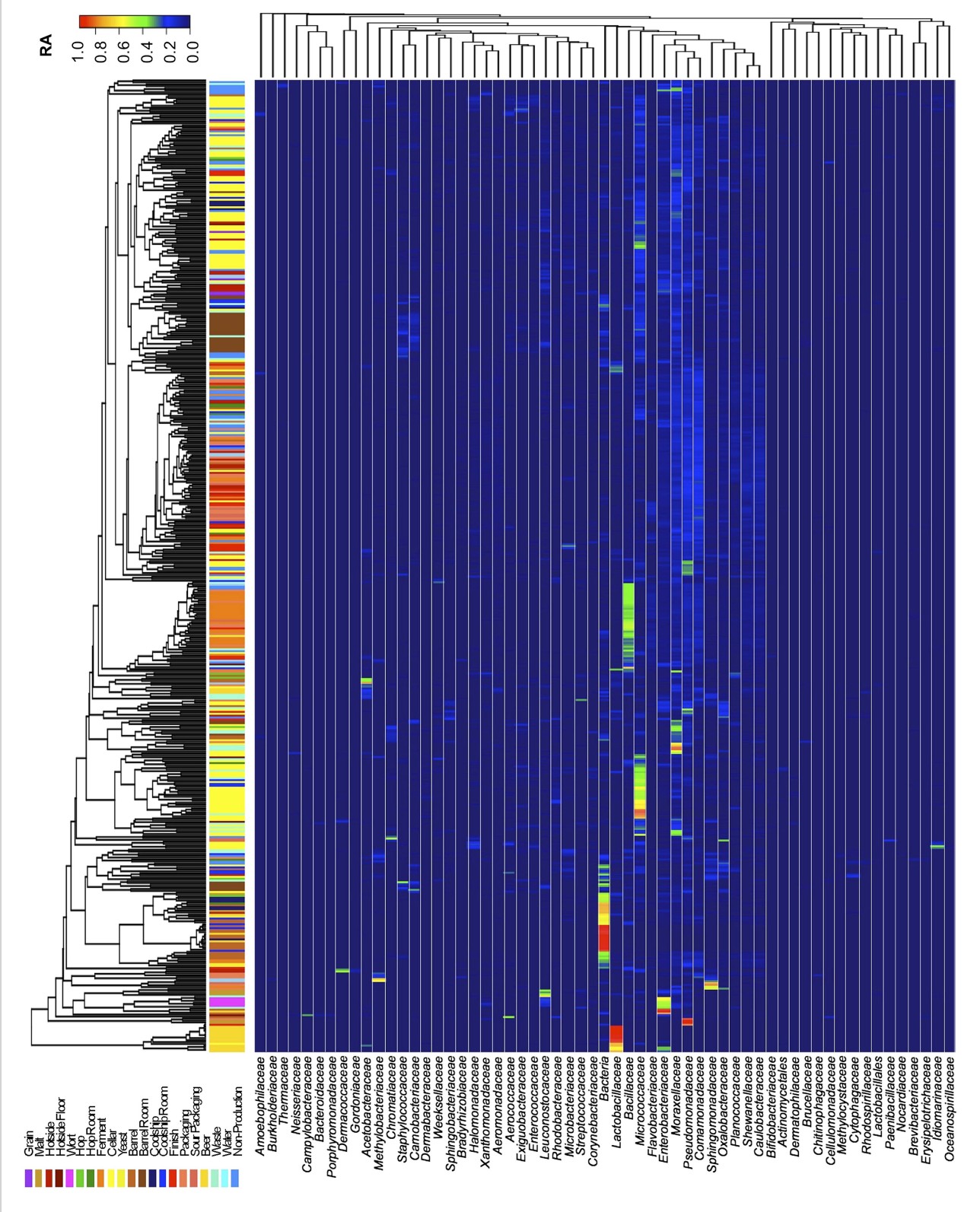

**Figure 3.** Taxon abundance heatmaps depict family-level relative abundance of bacteria across sampling sites detected by marker-gene sequencing. The relative abundances of each genus (columns) within each sample (rows) are indicated by the color of the intersecting tile. Sample types are indicated by colored bars to the left of each row, classified according to the location within the brewery (*Figure 1*) or the type of substrate (grain, wort, hops, beer). *Figure 3. continued on next page*

Figure 3. Continued

Dendrograms represent Bray–Curtis dissimilarity between samples (vertical trees) and shared-niche similarity between taxa (horizontal trees), respectively indicating taxonomic composition similarities and taxon co-occurrence patterns. Only taxa ≥0.05 relative abundance in at least one sample are shown.

a production brewery (indeed, hosing down floors is one obvious cause), increasing the likelihood for microbial spread and colonization from these surfaces onto other surfaces, including equipment. Human and insect traffic can also increase the rate of dispersal between these surfaces (e.g., from one room to another). Physical partitions appear to inhibit this passage as observed in this brewery and, where possible, may reduce the probability for contamination. This is particularly important in packaging areas, where microbial dispersal can directly contaminate sanitary surfaces and open packages, increasing the likelihood of spoilage in finished product. Separating packaging rooms from cellar and hotside operations and using local partitions, such as blast shields, may help protect packaging surfaces and beer from such routes of contamination. Further investigations into the relationship between building materials, spatial area, indoor conditions (e.g., airflow, humidity, and personnel traffic), and the rate and extent of microbial dispersion could yield important findings for optimal brewery design from the perspective of microbial control. These findings are instructive but indeed should not be interpreted with alarm: they illustrate microbial dispersal in a functioning brewery that rarely suffers from any cases of product contamination.

## Lactic acid bacteria profiles are also driven by substrate exposure

Different species of LAB are the principal spoilage bacteria in beer fermentations as well as important members of sour beer fermentations. However, short 16S rRNA gene amplicons are frequently inadequate to resolve reliable species-level identifications (*Bokulich and Mills, 2012a*). Therefore, we used LAB-TRFLP (*Bokulich and Mills, 2012a*) to characterize genus- and species-level LAB community compositions in a select subset of samples. This included raw material and beer samples, as well as surfaces on which detection of *Lactobacillales* by marker-gene sequencing was particularly high.

Results indicate that different surface and sample types exhibit distinct lactic acid bacterial patterns, corresponding to the substrates encountered at that site or contained in that sample (*Figure 8*). Wort samples contained a mixture of *Lactobacillus delbrueckii*, *Lactobacillus sakei*, *Lactobacillus hilgardii*, *Leuconostoc mesenteroides*, *Lactococcus lactis*, *Streptococcus* sp., and *Bacillus* sp. A, most of which were only rarely detected in other fermenting and bottled beer samples. Many of these species are also rarely found in beers (*Bokulich and Bamforth, 2013*), but instead appear associated with grain, hence their detection in wort. Coolship and fermenting sour beers (in this case coolship beers produced from different wort types) were dominated by *Pediococcus* and/or *L. lindneri*, corroborating previous studies of coolship beers in this brewery (*Bokulich et al., 2012b*). Fermenters and barrel surfaces that contacted these fermentations near the time of sampling exhibited similar communities, though *Lactobacillus brevis* and *Lactobacillus* sp. A were more common on these surfaces than in the beers or on other surfaces. Other sour and barrel-aged beers contained unique profiles, with involvement of other *Lactobacillus* species only weakly detected in coolship beers. Floor and packaging area surfaces contained a more diverse mixture of LAB, but primarily the *L. lindneri*, *L. brevis*, and *L. delbrueckii* detected in the wort and beer samples. Interestingly, only *Pediococcus* was detected on grain samples, though only weak amplification could be had from these samples, suggesting low LAB populations or inhibition of PCR by grain polyphenols, possibly suppressing the detection of less abundant populations. Hop pellet samples also contained a mixture of different LAB populations, including *Pediococcus*, *L. lindneri*, and *L. brevis*.

These results illustrate that substrate drives the composition of LAB communities as well as whole microbial communities and highlight the risk of cross-contamination between different equipment surfaces. The detection of *Lactobacillus* spp. on both filler heads (only one of which is used for sour beer packaging) makes this observation all the more pertinent. This observation is not likely to be an exceptional case; cross-contamination between processing areas is very likely the prevailing cause of spoilage in any brewery, where microbial biofilms have been previously reported even on packaging equipment (*Timke et al., 2005*; *Storgards et al., 2006*; *Timke et al., 2008*; *Mamvura et al., 2011*; *Matoulkova et al., 2012*). This observation underlines the need for constant hygiene surveillance in

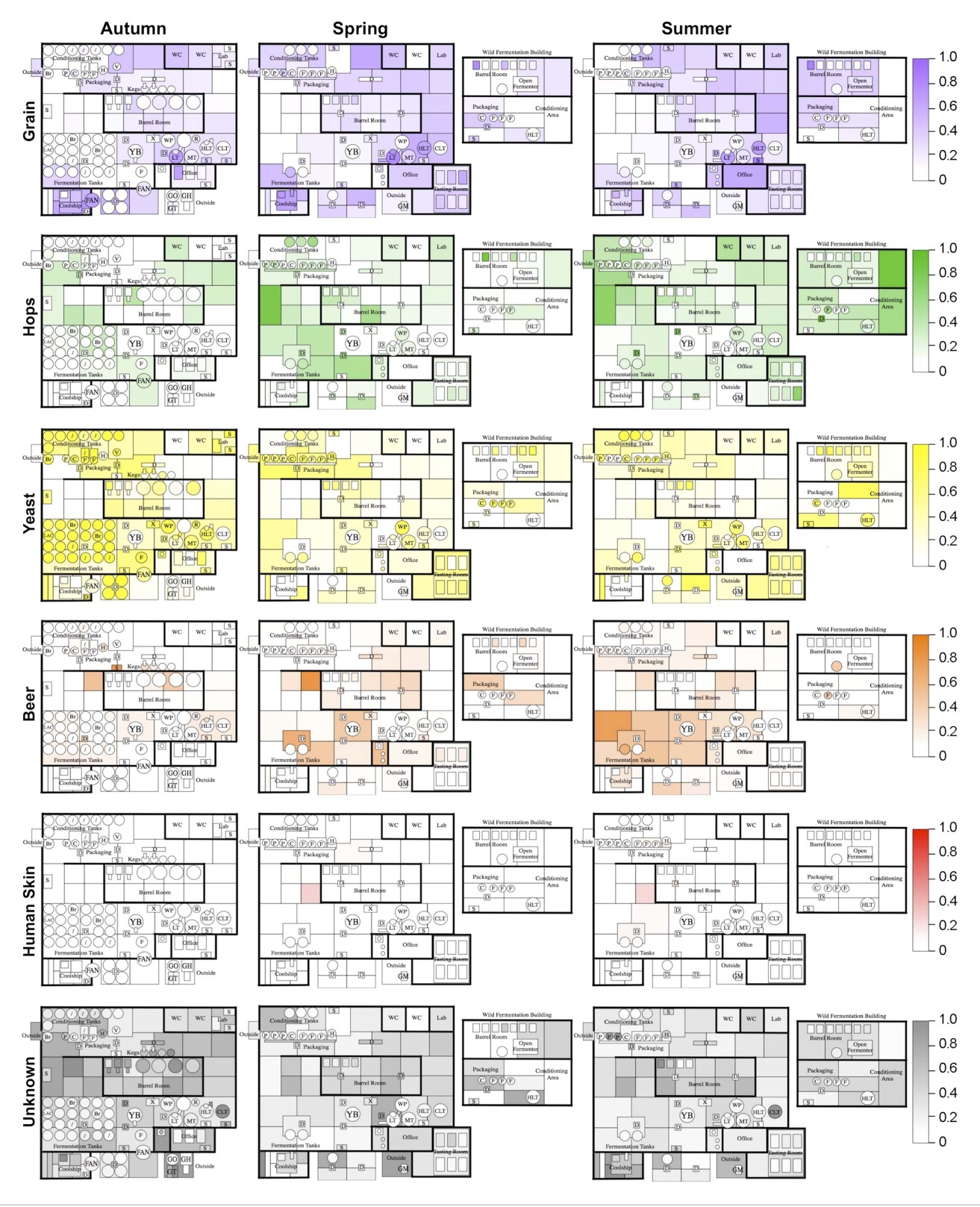

**Figure 4**. Mapping microbial contamination sources inside the brewery. Floorplans of brewery indicate the predicted relative contamination of brewery surfaces by microbial sources (grains, hops, yeast, beer, human skin, unknown) at each season, estimated by SourceTracker (*Knights et al., 2010*). Coloration of each surface indicates the relative degree of microbial contamination from that source type (as indicated by keys to the right of each row;

*Figure 4. continued on next page*

*Figure 4. Continued*

units are relative abundance). Contamination from outdoor air, soil, feces, freshwater, ocean water, and saliva was negligible (<0.001 relative abundance) and are not shown. See *Figure 1* for a floorplan key and description of surfaces.

breweries but is hardly cause for alarm, if the exceedingly low incidence of microbial spoilage in modern brewing practice is any indication.

## Beer contact predicts spoilage-gene distribution on brewery surfaces

Beer-spoiling LAB possess several mechanisms that support their growth and survival in beer (*Suzuki et al., 2006*). Several hop-resistance genes are principal among these, counteracting the antimicrobial effects of iso-a-acids (hop bittering resins) (*Simpson, 1993b*; *Simpson and Fernandez, 1994*). Hop-resistant LAB typically contain several of these genes, including *horA* (*Sami et al., 1997*; *Sakamoto et al., 2001*), *horB* (*Suzuki et al., 2005*; *Iijima et al., 2006*), *horC* (*Suzuki et al., 2005*; *Iijima et al., 2006*), and *hitA* (*Hayashi et al., 2001*), which are upregulated during growth in beer (*Bergsveinson et al., 2012*; *Pittet et al., 2013*). However, the presence of these genes within brewery environments and on brewing equipment has not been described previously. Thus, we sought to quantify the abundance of these beer-spoilage genes on brewing equipment and in beer during different times in conjunction with microbial community profiles.

Results demonstrate high gene frequencies on several surfaces within the brewery (*Figure 9*). Sour beer samples contained the highest counts, between $2.0 \times 10^4$–$4.8 \times 10^4$ copies/µl of *horC*, but fermenter and packaging area surfaces (filler heads, below bottling line belt, and packaging sink) also registered between $2.8$–$7.8 \times 10^2$ copies/cm². None of these alleles were detected on hop samples, keg samples, or one barrel bung (stopper) sample, though $1.1 \times 10^3$ total copies/cm² were detected on a keg faucet used for attaching kegs to beer lines at the brewery. Among the genes analyzed, *horC* was the most abundant (*Figure 9*), which is interesting when considered in the context of previous work showing that presence of this gene correlates with increased hop-tolerance and beer-spoilage ability (*Fujii et al., 2005*; *Iijima et al., 2006*; *Bergsveinson et al., 2012*) and the plasmid carrying *horC* is the most important for supporting growth of *Lactobacillus brevis* in beer (*Bergsveinson et al., 2015*). The preferential expression of this gene observed in these previous studies and the relative increased abundance with which it was found in this study suggests horC is an important gene for facilitating beer-spoilage and is consequently selected for in the brewery environment, particularly in areas where sour beers are produced. The purported transcriptional regulator of *horC*, *horB*, was detected at stable ratios relative to *horC* in all samples, supporting this putative function (*Iijima et al., 2006*). The least frequently observed hop-resistance gene, *hitA*, is involved in manganese transport (*Hayashi et al., 2001*), supporting resistance against manganese depletion by iso-a-acids (*Behr and Vogel, 2010*). Other studies have observed similarly low frequencies of *hitA* presence and expression in LAB relative to the other hop-resistance genes (*Haakensen and Ziola, 2008*; *Bergsveinson et al., 2012*).

The spoilage genes *horA, horB,* and *horC* all display high degrees of intercorrelation (Pearson's $r = 0.83$–$1.0$, $p < 0.01$) and significant but lesser correlation to *hitA* ($r = 0.48$–$0.64$, $p \leq 0.04$). All spoilage genes except for *hitA* demonstrate significant correlation with bulk detection of *Lactobacillales* via 16S rRNA gene sequencing ($r = 0.53$–$0.74$, $p \leq 0.03$), while bulk *Lactobacillales* and all spoilage genes but *horA* correlate significantly with *L. lindneri* detection via LAB-TRFLP ($r = 0.48$–$0.77$, $p \leq 0.04$). The only gene correlated with *Pediococcus* abundance via LAB-TRFLP was *horA* ($r = 0.57$, $p = 0.01$), consistent with previous observations that *horA* is the primary known hop-resistance gene observed in *Pediococcus* spp. (*Haakensen and Ziola, 2008*). Interestingly, no resistance genes correlated with *L. brevis*, strains of which are among the most common brewery contaminants and most commonly positive for hop-resistance genes (*Haakensen and Ziola, 2008*). This likely reflects the strains detected in this brewery only, and *L. brevis* was only a minor constituent of sour beers and processing surfaces (*Figure 7*).

These results are the first indication of hop-resistance-gene abundance within a brewery environment. As the primary reservoir for spoilage microbes in beer production, tracking spoilage genes on brewing surfaces and materials is important for understanding contamination risks arising from the environment. Sour beer contained the highest gene abundance, which is predictable given

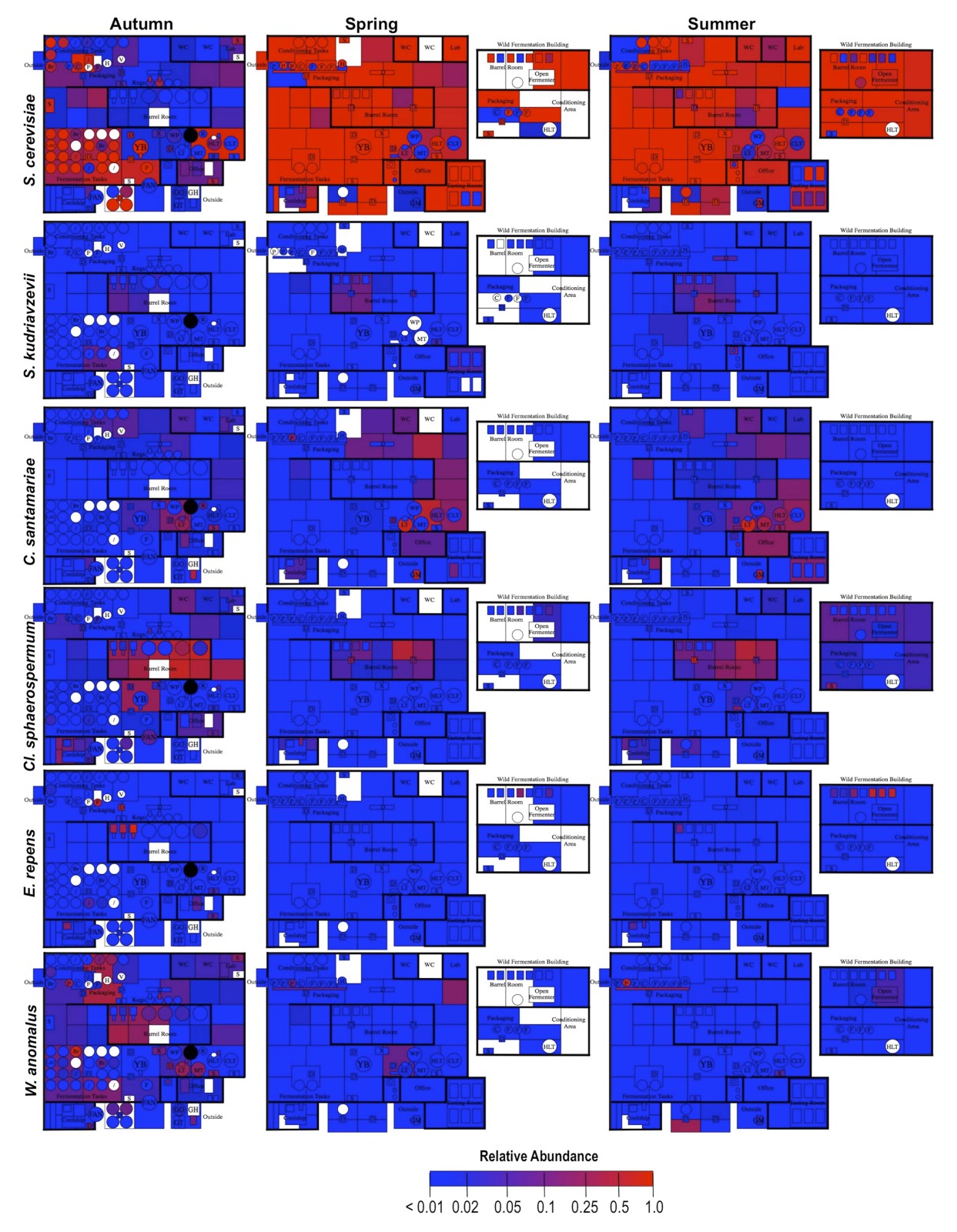

**Figure 5**. Spatial distribution heatmaps of fungi in brewery environments across seasons. Plots indicate relative abundance of fungal taxa detected by ITS sequence reads across brewery surfaces at different times: Autumn (left), Spring (center), and Summer (right). See *Figure 1* for a floorplan key and description of *Figure 5. continued on next page*

Figure 5. Continued

surfaces. Note that the floorplans change between seasons as some samples were only collected as specific timepoints and the wild brewing facility was built and opened during the Spring sampling time.

that the LAB growing in these (coolship) beers must be adapted to resist hop antimicrobials. Likewise, detection on barrel surfaces is also expected, given their regular and direct contact with these sour beers. Fermenters and packaging-line surfaces also exhibited fairly high levels of hop-resistance genes, with the highest detection on surfaces that contact sour beers and unsanitary surfaces (below packaging belt, packaging-line sink). The only sanitary, conventional beer-making surface among these is the second filler head, which only exhibited 4 cumulative copies/cm$^2$ of the resistance genes, compared to $2.1 \times 10^2$ in the first filler head, which is used for packaging sour and barrel-aged beers. Nevertheless, this second filler head is also a sanitary surface ready for packaging, and should be free of contaminants. DNA was used for ddPCR and thus these counts may represent dead cells present on the equipment surfaces; the relative proportion of viable cells is unknown. If the resistance genes detected by ddPCR do in fact represent any viable cells, this demonstrates the resilience of these bacterial populations on equipment surfaces that regularly contact 'contaminated' (in this case, intentionally contaminated) beers. These findings argue for careful separation of equipment used in conventional beer-making from that used for sour beers: equipment that contacts fermenting and finished sour beer or barrels, including pumps, hoses, and especially packaging equipment, is best dedicated to sour beer production and should not be used to handle conventional beers.

No hop-resistance genes were detected on hop pellets, though LAB were present, indicating that hops are probably not a significant source of beer-spoilage bacteria. Iso-alpha-acids, the antimicrobial compounds released from hops and against which hop-resistance genes confer protection, are generated by the breakdown of humulones during boiling, and are not present in raw hops (*Steenackers et al., 2015*). Undissociated humulones are significantly less inhibitory to LAB, and the antimicrobial effects of hops are dependent upon acidic conditions (*Simpson and Smith, 1992*). Thus, the selective pressure to acquire and maintain hop-resistance genes only exists in the presence of hopped beer, not in raw hops. Only two hop pellets were tested for hop-resistance genes, but this finding suggests that the brewery environment itself is the site of hop-resistance-gene propagation. Larger studies of hop-resistance genes on brewery surfaces and brewing materials will illuminate the role of environmental vs raw material contamination in hop-resistance-gene transfer, and sites, conditions, and mechanisms of transfer within the brewery environment.

Tracking spoilage-gene flow across brewery surfaces presents a unique opportunity for understanding spoilage dynamics within food-processing systems and other built environments in general. This approach will facilitate understanding how spoilage resistance propagates within production environments and the reservoirs and vectors encouraging its spread, yielding novel insight for controlling spoilage—focused on gene transmission rather than taxonomic populations. Moving forward, this facility-surveillance model opens many questions and possibilities for mapping microbial spoilage dynamics within food-production systems. What relationship do indoor environmental factors, building design, and surface materials have with microbial transmission? What roles do cleaning and other processes play in controlling contamination on a facility-wide scale? What other biomarkers are associated with contamination and how does the spoilage-allele landscape behave over time and in response to these conditions? The potential advantages of studying microbiology of breweries are not confined to food systems alone, and lessons learned here may aid our understanding of microbial communities in other built environments. For example, the hop-resistance genes include ABC multi-drug transporters similar to other antimicrobial-resistance genes (*Sakamoto et al., 2001*). Studying and manipulating their transmission within breweries may aid understanding of similar gene-transfer events involved in pathogenesis in hospitals, homes, water systems, or public environments where in situ modeling of such genes is unfeasible or a potential health hazard. Breweries are a useful model for testing general theories of microbial transmission and spoilage-gene dispersal in situ for functional indoor environments and food systems, as beer spoilage is not actually detrimental to human health—merely human pleasure.

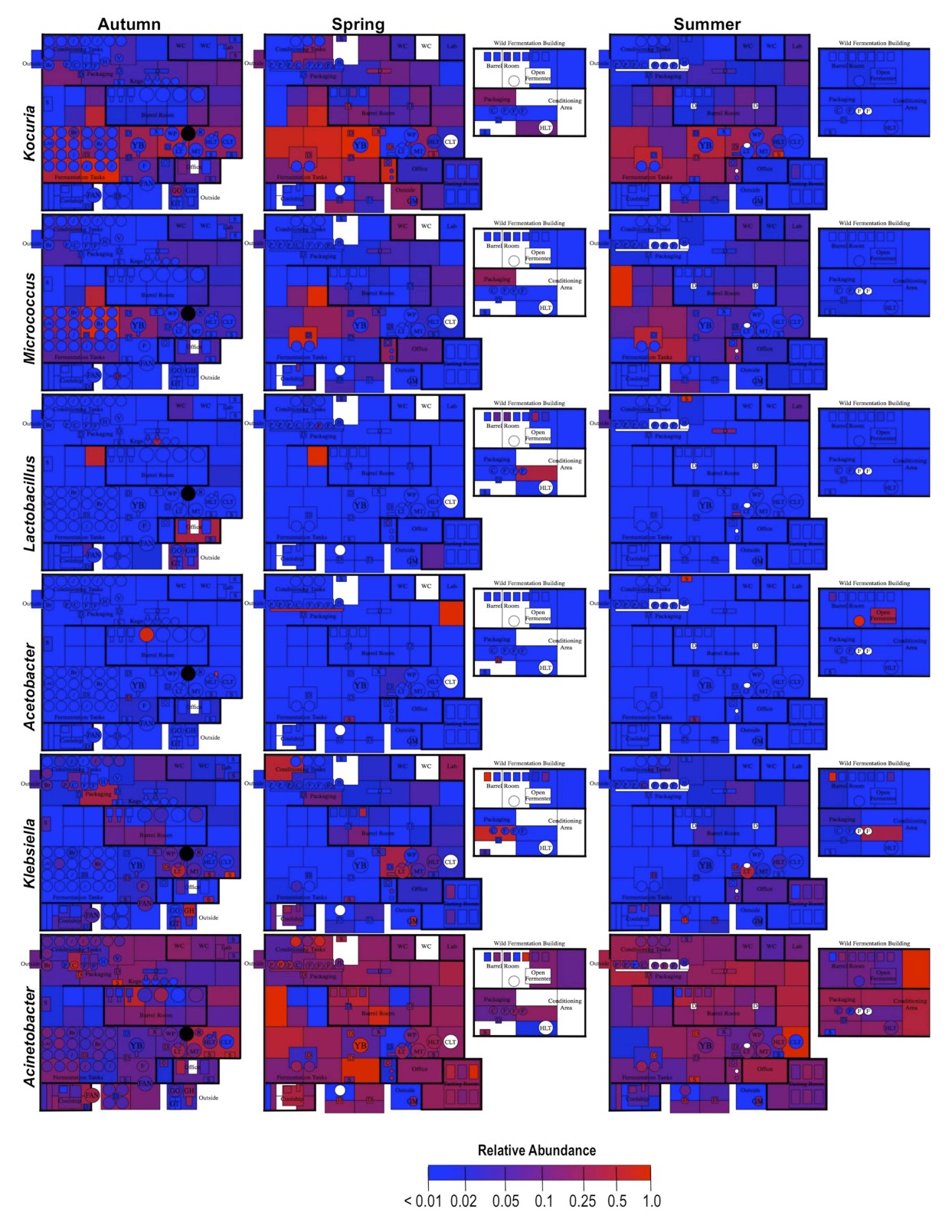

**Figure 6**. Spatial distribution heatmaps of bacteria in brewery environments across seasons. Plots indicate relative abundance of bacterial taxa detected by 16S rRNA gene sequence reads across brewery surfaces at different times: Autumn (left), Spring (center), and Summer (right). Scales on right represent relative abundance scale (maximum 1.0) for each row of plots. See *Figure 1* for a floorplan key and description of surfaces. Note that the floorplans
*Figure 6. continued on next page*

*Figure 6. Continued*
change between seasons as some samples were only collected as specific timepoints and the wild brewing facility was built and opened during the Spring sampling time.

## Materials and methods

### Facility description

Samples were collected from a single brewery located in North America. This brewery operates as a conventional brewing facility but also contains a dedicated coolship room, barrel room, and a secondary cellar building, where it produces coolship and other sour and barrel-aged beers seasonally. The coolship beers (a type of sour beer produced without inoculation) are produced following the classic Belgian tradition, and their microbial profiles in this brewery have been investigated previously (*Bokulich et al., 2012b*). The other sour beers are produced using pure-culture inocula of non-*Saccharomyces* yeasts and bacteria, but the term 'sour beer' refers to both coolship and other sour beers elsewhere in this study unless if a distinction is made between these types of beer. The secondary cellar building was built and opened during the course of the collection period. Thus, during the first sample collection time point (Fall, 2012) sour beers were fermented and stored in the barrel room of the main facility. For the remaining times (Spring and Summer, 2013), all sour beers were held in this second 'wild fermentation' facility. A facility map and flow–through diagrams relating brewing stages to this space are presented in *Figure 1*.

### Sample collection and DNA extraction

Samples were collected in Fall 2012, Spring 2013, and Summer 2013. In all, 445 surface swabs and 56 beer and ingredient samples were collected (*Figure 1*) as described previously (*Bokulich et al., 2013b*). DNA was extracted using the ZR-96 Fecal DNA MiniPrep Kit (Zymo Research, Irvine, CA), with bead beating in a FastPrep-24 bead beater (MP Bio, Solon, OH), and stored at −20°C until further processing.

### Sequencing library construction

Amplification and sequencing were performed as described previously for bacterial (*Bokulich, 2012b*) and fungal communities (*Bokulich and Mills, 2013b*). The V4 domain of bacterial 16S rRNA genes was amplified using primers F515 (5′–*NNNNNNNN***GT**GTGCCAGCMGCCGCGGTAA–3′) and R806 (5′–GGACTACHVGGGTWTCTAAT–3′) (*Caporaso et al., 2011*), with a unique 8 nt barcode (italicized poly-N section) and 2 nt linker sequence (bold) at the 5′ terminus. Fungal internal transcribed spacer (ITS) 1 loci were amplified with primers BITS (5′–*NNNNNNNN***CT**ACCTGCGGARGGATCA–3′) and B58S3 (5′–GAGATCCRTTGYTRAAAGTT–3′) (*Bokulich and Mills, 2013b*). Amplicons were combined into two separate pooled samples (keeping bacterial and fungal amplicons separate) at roughly equal amplification intensity ratios, purified using the Qiaquick spin kit (Qiagen, Germantown, MD), and submitted to the UC Davis DNA Technologies Core for Illumina paired-end library preparation, cluster generation, and 250 bp paired-end sequencing on an Illumina MiSeq instrument in four separate runs (separating bacterial and fungal libraries).

### Data analysis

Raw fastq files were demultiplexed, quality-filtered, and analyzed using QIIME v.1.7.0 (*Caporaso et al., 2010b*). The 250-bp reads were truncated at any site of ≥3 sequential bases receiving a quality score < Q10, and any read with <75% (of total read length) consecutive high-quality base calls was discarded (*Bokulich et al., 2013c*). Operational taxonomic units (OTUs) were clustered at 97% identity using QIIME's open-reference OTU-picking pipeline using UCLUST-ref (*Edgar, 2010*) against either the Greengenes 16S rRNA gene database (May 2013 release) (*McDonald et al., 2012*) or the UNITE fungal ITS database (*Abarenkov et al., 2010*), modified as described previously (*Bokulich and Mills, 2013b*). OTUs were classified taxonomically using RDP classifier (*Wang et al., 2007*) for bacteria and BLAST (*Altschul et al., 1990*) for fungi. Any OTU comprising less than 0.0001% of total sequences for each run were removed (*Bokulich et al., 2013c*). Bacterial 16S rRNA gene sequences were aligned

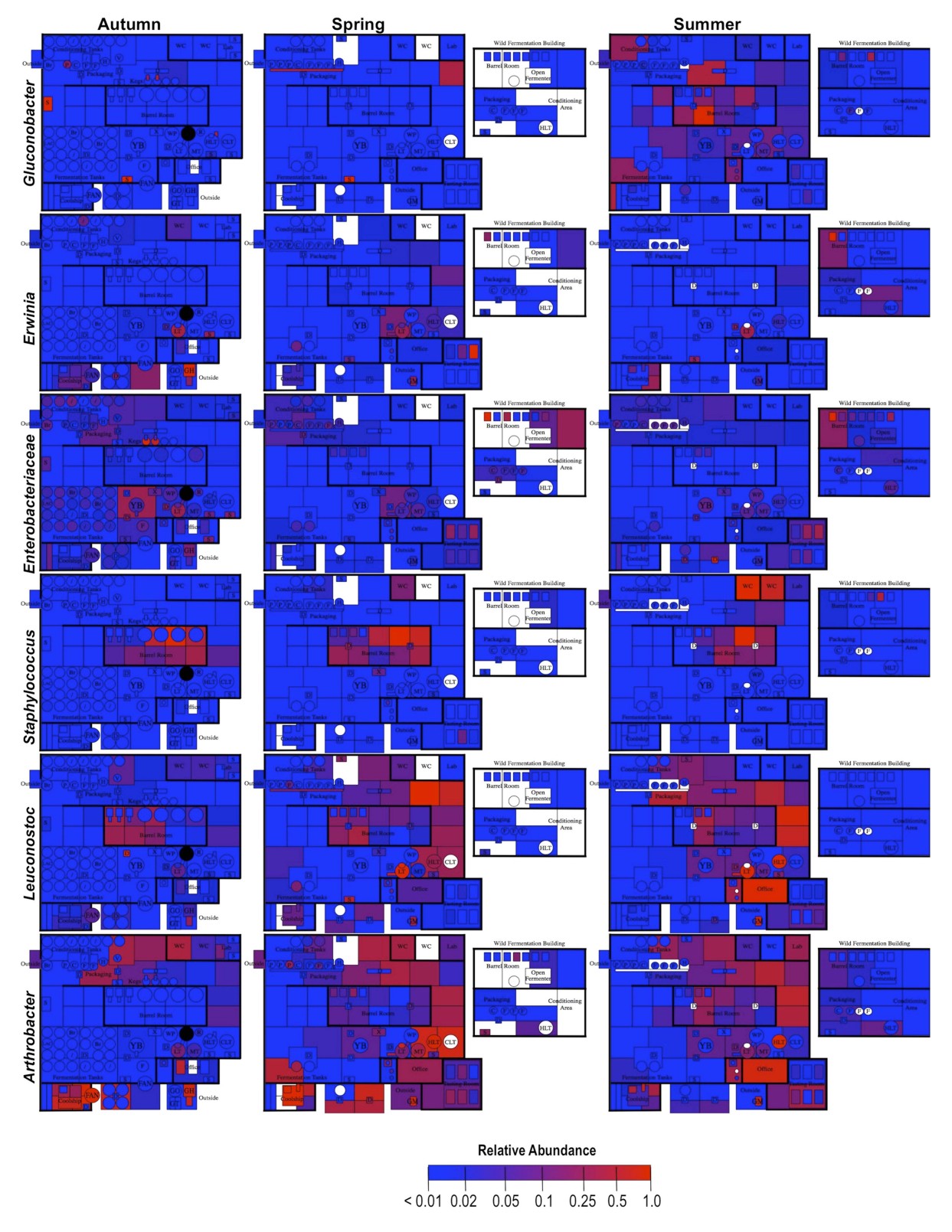

**Figure 7**. Spatial distribution heatmaps of bacteria in brewery environments across seasons (part 2). Plots indicate relative abundance of bacterial taxa detected by 16S rRNA gene sequence reads across brewery surfaces at different times: Autumn (left), Spring (center), and Summer (right). Scales on right
*Figure 7. continued on next page*

Figure 7. Continued
represent relative abundance scale (maximum 1.0) for each row of plots. See *Figure 1* for a floorplan key and description of surfaces. Note that the floorplans change between seasons as some samples were only collected as specific timepoints and the wild brewing facility was built and opened during the Spring sampling time.

using PyNAST (*Caporaso et al., 2010a*) and chimeric sequences were identified using ChimeraSlayer (*Haas et al., 2011*). Sequences failing alignment or identified as chimera were removed prior to downstream analysis. OTU tables were evenly subsampled to 400 sequences per sample for all statistical tests. MANOVA with 999 permutations was used to test significant difference between sample categories based on Bray–Curtis distance using Adonis (*Anderson, 2001*). Kruskal–Wallis tests were used to identify significantly discriminant taxa (with Bonferroni error correction) between sample groups. Pearson product-moment correlation analyses were performed using R software. Environmental surveillance heatmaps were generated based on taxonomic abundance tables generated in QIIME and visualized using SitePainter 1.1 (*Gonzalez et al., 2012*). Bacterial OTU source-sink relationships were tested using SourceTracker (*Knights et al., 2010*) with 1000 burn-ins, 25 restarts, and rarefaction to 100 OTUs. Source-tracking predictions used bacterial profiles of samples collected in this study, coolship beers from this brewery analyzed in a previous study (*Bokulich et al., 2012b*), and outdoor air (*Bowers et al., 2012*, *2013*), soil, saliva, feces, human skin, freshwater, and ocean water samples (*Caporaso et al., 2011*, *2012*) from previously published studies as source samples. All studies were performed using bacterial V4 16S rRNA with the same F515/R806 primer pair.

## Terminal restriction fragment length polymorphism (TRFLP)

Lactic acid bacteria-specific TRFLP was performed as described previously using the primers NLAB2F (5′-(5HEX)-GGCGGCGTGCCTAATACATGCAAGT-3′) and WLAB1R (5′-TCGCTTTACGCC-CAATAAATCCGGA-3′) (*Bokulich and Mills, 2012a*). Purified amplicons were digested using enzymes MseI and Hpy188I and submitted to the UC Davis College of Biological Sciences Sequencing Facility for capillary electrophoresis fragment separation. Electropherogram traces were visualized using the program Peak Scanner v1.0 (Applied Biosystems, Carlsbad, CA) using

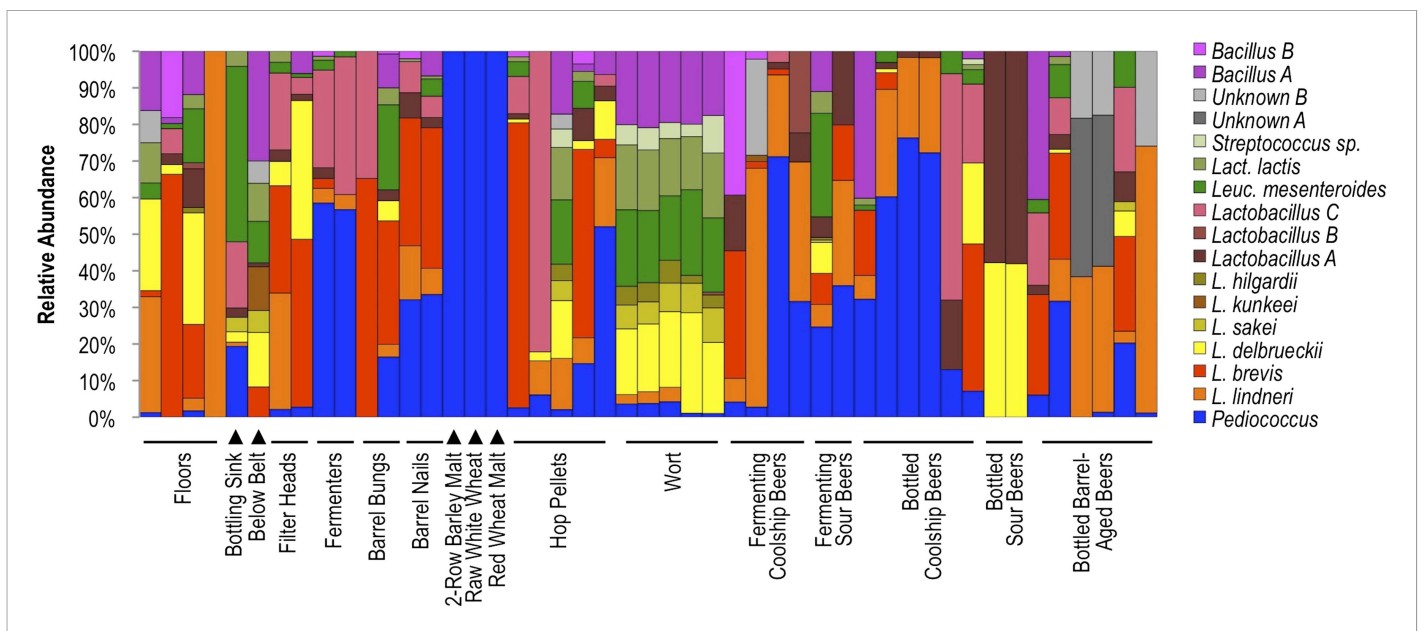

**Figure 8**. Lactic acid bacterial community composition on brewery surfaces, beers, and ingredients. LAB-TRFLP profiles of samples exhibiting high *Lactobacillales* relative abundance by 16S rRNA gene sequencing.

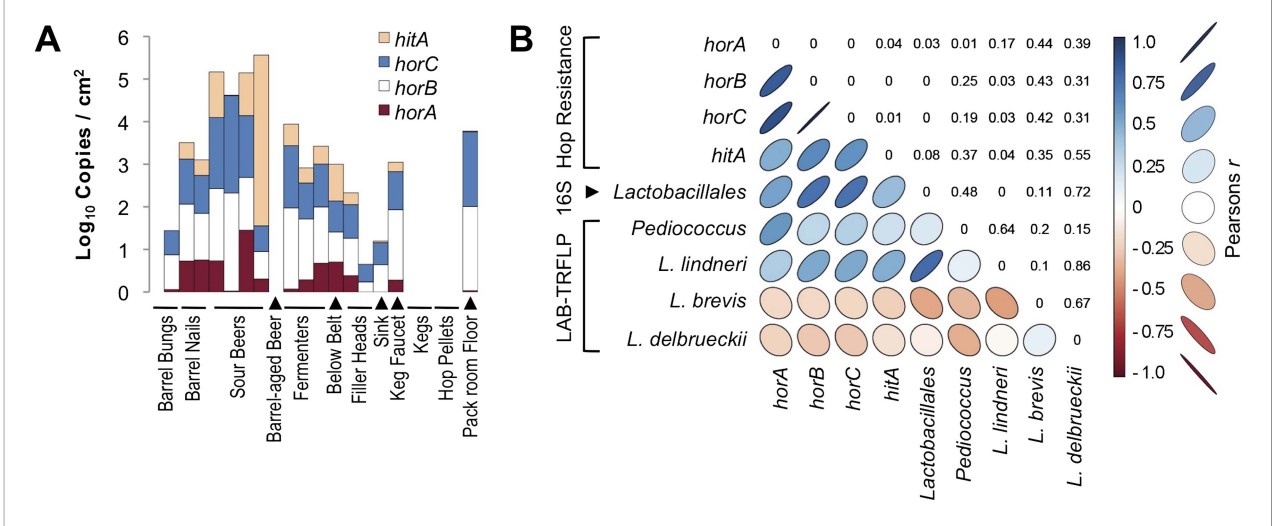

**Figure 9**. Hop-resistance gene frequency on brewery surfaces and beers. (**A**) ddPCR detection of *hitA, horA, horB,* and *horC* on surfaces (detected as copies/cm²) and in beers (copies/ml). Bar height indicates cumulative log gene abundance; colors indicate relative gene frequencies superimposed on these bars. Two barrel bung (stopper) samples are depicted on the left, one has no detection. (**B**) Pearson product-moment correlation matrix between hop-resistance genes, *Lactobacillales* relative abundance by 16S rRNA gene sequencing, and relative abundance of the dominant lactic acid bacteria detected by LAB-TRFLP. The color and shape of correlation ellipses (lower-left) indicate Pearson's product-moment correlation coefficient (*r*) between intersecting variables, as depicted in the key to the right. Correlations with larger positive *r* values are depicted as darker blue with increasingly narrow, upward-pointing ellipses. Correlations with larger negative *r* values are depicted as darker red with increasingly narrow, downward-pointing ellipses. Weaker correlations are depicted as wider, lighter colored ellipses. The corresponding p values for all correlation tests are provided in the reflected intersection (top-right).

a baseline detection value of 10 fluorescence units. Peak filtration and clustering were performed with R software using IBEST TRFLP tools (*Abdo et al., 2006*). OTUs were identified based on an empirical TRFLP database (*Bokulich and Mills, 2012a*) and an in silico digest database generated with MiCA (*Shyu et al., 2007*) of good-quality 16S rRNA gene sequences from RDP (*Cole et al., 2014*), allowing up to three nucleotide mismatches within 15 bp of the 5′ terminus of the forward primer.

## Droplet digital PCR

In order to quantify hop-resistance gene abundance on brewery surfaces, droplet digital PCR (ddPCR) was used to enumerate the genes *horA, horB, horC,* and *hitA*. ddPCR was performed using the QX100 Droplet Digital PCR setup and protocol (Bio-Rad). ddPCR was performed in 20-µl reactions containing 3 ng/µl of DNA template, 900 nmol of each primer, 250 nM of each probe, and 1× Bio-Rad Droplet PCR Supermix. Primer and probe sequences, melting temperatures, 5′ fluorophore probe labels, and amplicon lengths for each gene target are shown in *Table 1*. All probes contained a 3′ IowaBlack FQ quencher paired with different 5′ fluorescent labels (*Table 1*). Each 20 µl reaction was then pipetted into separate wells of a disposable eight channel droplet generation cartridge (Bio-Rad) and 70 µl of droplet generation oil (Bio-Rad) loaded into the cartridge oil wells. The cartridge was then inserted into the QX100 droplet generator (Bio-Rad) and each sample was portioned into droplet-sized water-in-oil emulsions. Following droplet generation, emulsion samples were transferred to a 96-well PCR plate (Eppendorf). The plate was then hot-sealed with foil cover and subjected to conventional PCR in the CFX96 Touch Real-Time PCR (Bio-Rad). Thermal cycling conditions consisted of an activation period for 10 min at 95°C, followed by 40 cycles of a denaturation step for 30 s at 94°C, and an annealing-extension step for 60 s at the optimal annealing temperature (59–59.6°C), using a ramp rate of 2.5°C/s for each step and a final inactivation step of 98°C for 10 min. After PCR amplification, the plate was loaded into the QX100 Droplet Digitial PCR (Bio-Rad) and analyzed for absolute signal quantification of each fluorescence channel in each well. Signal detection and data processing were performed using Quantasoft Software v.1.3.2 (Bio-Rad).

**Table 1**. Hop-Resistance Gene Primers and Probes for ddPCR

| Target | Tm* | 5′ Label | Probe | Forward primer | Reverse primer | bp† |
|--------|-----|----------|-------|----------------|----------------|-----|
| horB | 59 | FAM | TCGCGGCCAAGTGATACTTATCCTGA | AGTCGACACAAAATCCTGAATCA | AGCCTTGATCAATCGTCAGAC | 88 |
| hitA | 59 | HEX | ACAGAATAACGGCAACCAGTGTCGCAA | TCCTGTTGCTTCTGATGAAATTGG | CCGCTAAGAATACTTCGTAGGTGA | 105 |
| horA | 59.6 | FAM | CGCCGTTCCGCTCGTCTTGATCTGCC | TGGACTGGCGGATGACTATC | CTGTCTCGCTCTGGCAAC | 104 |
| horC | 59.6 | HEX | ACCACGCCAATGCCACTAGAAGCATGG | ACACGGTTAATGGCACAGC | GTTCGCGCCATAAAATAAGAGAGG | 87 |

*Tm = melt temperature (°C).
†Nucleotide length (bp).

## Data availability

All raw marker-gene sequencing data are publicly deposited in QIITA (http://qiita.colorado.edu/) under the accession number 10105 (http://qiita.colorado.edu/study/description/10105).

## Acknowledgements

The authors thank the brewers involved in this study for generously donating their time, beer, and support with sampling efforts. Chad Masarweh and Morgan Lee are thanked for their technical support. The authors thank Gail Ackermann for helping review metadata and depositing raw data in QIITA database. NAB was supported by the Samuel Adams Scholarship Fund, Brian Williams Scholarship Fund (both awarded by the American Society of Brewing Chemists Foundation), the John E Kinsella Memorial Award, the American Wine Society Educational Foundation Endowment Fund scholarship, an American Society for Enology and Viticulture scholarship, and Grant Number T32-GM008799 from NIGMS-NIH during the completion of this work. JB was supported during this work by Graduate Scholarships from the Colleges of Medicine, and Graduate Studies and Research at the University of Saskatchewan; by the 2012 and 2013 Ecolab Scholarships (awarded by the American Society of Brewing Chemists Foundation); and by Discovery Grant 24067 (awarded to BZ) from the Natural Sciences and Engineering Research Council of Canada.

## Additional information

### Funding

| Funder | Author |
|--------|--------|
| National Institute of General Medical Sciences (NIGMS) | Nicholas A Bokulich |
| Natural Sciences and Engineering Research Council of Canada | Barry Ziola |

The funders had no role in study design, data collection and interpretation, or the decision to submit the work for publication.

### Author contributions

NAB, Conception and design, Acquisition of data, Analysis and interpretation of data, Drafting or revising the article; JB, BZ, Acquisition of data, Analysis and interpretation of data, Drafting or revising the article; DAM, Conception and design, Drafting or revising the article

### Author ORCIDs

Barry Ziola, http://orcid.org/0000-0003-4836-7576

## Additional files

### Major datasets

The following datasets were generated:

| Author(s) | Year | Dataset title | Dataset ID and/or URL | Database, license, and accessibility information |
|---|---|---|---|---|
| Bokulich NA | 2015 | Bokulich_brewery_surfaces | 10105; http://qiita.colorado.edu/study/description/10105 | Available on login at Qiita (http://qiita.colorado.edu/). |

The following previously published dataset was used:

| Author(s) | Year | Dataset title | Dataset ID and/or URL | Database, license, and accessibility information |
|---|---|---|---|---|
| Bokulich NA | 2013 | bokulich_quality_dataset9 | 1689; http://qiita.colorado.edu/study/description/1689 | Available on login at Qiita (http://qiita.colorado.edu/). |

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
