## [Decision Letter]

Thank you for sending your work entitled “Mapping microbial ecosystems and spoilage-gene flow in breweries highlights patterns of contamination and resistance” for consideration at *eLife*. Your article has been favorably evaluated by Ian Baldwin (Senior editor), Roberto Kolter (Reviewing editor), and two outside reviewers, one of whom was Greg Caporaso.

The Reviewing editor and the other reviewers discussed their comments before we reached this decision, and the Reviewing editor has assembled the following comments to help you prepare a revised submission.

The authors present a nice study of tracking the source of microbes and antimicrobial resistance genes through breweries. The analysis is well done. While overall we deemed the manuscript presented a very nice story that could be of broad interest, there were several concerns that should be addressed in revising the submission.

First, we were concerned with the data and interpretations of causative sources of contamination based on the Sourcetracker software. Given that the results are a prediction of possible contamination, it will be helpful to know what the confidence in these predictions is. In addition, how do the authors distinguish between alternate possibilities such as 1) microbes coming instead from untested sources, 2) source and sample just happening to have similar communities, or 3) both source and sample being “contaminated” by another untested source. In other words, how likely is it that the listed source is actually causative of the “contamination”? This seems particularly difficult to determine in the coolship analysis. One dominant source is the beer-packaging surface (Figure 4). Isn't it equally likely that the beer itself was the source of contamination of the packaging surface? Also, is the barrel “source” the fermenting beer inside the barrel, the interior surface of an empty barrel (have the barrels been used for brewing before or are they new?), or the exterior of the barrel? Again, how can the “source” here really be interpreted? Some additional clarification would be very helpful. In short, the authors should elaborate on what other likely sources there are, and test these as possible sources, which might provide more information for the sites “influenced by unknown sources” (third paragraph of the Results and Discussion section). Such sources could include human skin and saliva, as well as rinse water, soil and outdoor air, and could easily be included analysis. Given that the authors used the 515F/806R primers for bacterial sequencing, it should be straight-forward to include samples from other studies for comparisons.

Second, the QIIME Database is currently offline, so the authors should deposit to an additional repository, and/or ensure that their samples are in the new version of the database (QiiTA) when it comes online. As far as we understand, this should happen soon or might already have happened. In the meantime, if QiiTA is not up and running, the sequences should be deposited in a repository such as EBI. That way, when QiiTA eventually comes online, a transfer should be easy.

Finally, in the Abstract the authors mention that the concepts learned from breweries are readily translatable to other built environments. We think the authors should come back to this point in the Discussion section, as right now it's not clear how these concepts translate to other, non-food-production environments. For example, can their findings provide insight into contamination sources in hospitals, or are the building systems too different? This would greatly add to the general interest of the manuscript.

---

## [Author Response]

*First, we were concerned with the data and interpretations of causative sources of contamination based on the Sourcetracker software. Given that the results are a prediction of possible contamination, it will be helpful to know what the confidence in these predictions is. In addition, how do the authors distinguish between alternate possibilities such as 1) microbes coming instead from untested sources, 2) source and sample just happening to have similar communities, or 3) both source and sample being “contaminated” by another untested source. In other words, how likely is it that the listed source is actually causative of the “contamination”? This seems particularly difficult to determine in the coolship analysis. One dominant source is the beer-packaging surface (*Figure 4*). Isn't it equally likely that the beer itself was the source of contamination of the packaging surface? Also, is the barrel “source” the fermenting beer inside the barrel, the interior surface of an empty barrel (have the barrels been used for brewing before or are they new?), or the exterior of the barrel? Again, how can the “source” here really be interpreted? Some additional clarification would be very helpful. In short, the authors should elaborate on what other likely sources there are, and test these as possible sources, which might provide more information for the sites “influenced by unknown sources” (third paragraph of the Results and Discussion section). Such sources could include human skin and saliva, as well as rinse water, soil and outdoor air, and could easily be included analysis. Given that the authors used the 515F/806R primers for bacterial sequencing, it should be straight-forward to include samples from other studies for comparisons*.

Thank you for these comments. We have revised the manuscript to clarify that sourcetracker analysis is not determining causality of contamination sources. The sourcetracker analysis has been re-done incorporating other source samples from published studies, as suggested. These include human skin, outdoor air, human saliva, feces, soil, freshwater, and ocean water from a number of other studies. We have also included beer samples as possible sources. This analysis revealed that human skin and beer were sources of contamination on some surfaces, but other additional sources were negligible (< 0.1% relative contribution). Otherwise, results for other contaminant sources are similar to the original analysis. These results are shown in a new Figure 4.

As for the coolship analysis, we have removed this, as we agree with the reviewers’ point that it is difficult to determine the exact nature of the source/sink relationship with these samples.

*Second, the QIIME Database is currently offline, so the authors should deposit to an additional repository, and/or ensure that their samples are in the new version of the database (QiiTA) when it comes online. As far as we understand, this should happen soon or might already have happened. In the meantime, if QiiTA is not up and running, the sequences should be deposited in a repository such as EBI. That way, when QiiTA eventually comes online, a transfer should be easy*.

We are fully committed to making our data publicly available. However, our preferred database, QIITA (http://qiita.colorado.edu/), is still in alpha release. We are working with the database curators to finalize the data upload and ensure compliance with data standards, as this system is new. However we believe this database is the best suited for the type of data, as it will support streamlined meta-analyses for other built environment projects deposited in that database, facilitating future analyses. Data accessibility information is provided in the manuscript, but we are working on finalizing the publication of these data in QIITA.

*Finally, in the Abstract the authors mention that the concepts learned from breweries are readily translatable to other built environments. We think the authors should come back to this point in the Discussion section, as right now it's not clear how these concepts translate to other, non-food-production environments. For example, can their findings provide insight into contamination sources in hospitals, or are the building systems too different? This would greatly add to the general interest of the manuscript*.

This is a very good point. We believe that lessons learned in breweries could translate to other non-food built environments, including hospitals. We have discussed some ideas in the thirteenth paragraph of the Results and Discussion section.